# Potential Health Risks of Lead Exposure from Early Life through Later Life: Implications for Public Health Education

**DOI:** 10.3390/ijerph192316006

**Published:** 2022-11-30

**Authors:** Adejoke Christianah Olufemi, Andile Mji, Murembiwa Stanley Mukhola

**Affiliations:** Office of the Dean, Faculty of Humanities, Tshwane University of Technology, Private Bag X680, Pretoria 0001, South Africa

**Keywords:** children, heavy metals, health risks, human health, intelligence quotient, lead exposure

## Abstract

Lead (Pb) exposure has been a serious environmental and public health problem throughout the world over the years. The major sources of lead in the past were paint and gasoline before they were phased out due to its toxicity. Meanwhile, people continue to be exposed to lead from time to time through many other sources such as water, food, soil and air. Lead exposure from these sources could have detrimental effects on human health, especially in children. UNICEF reported that approximately 800 million children have blood lead levels (BLLs) at or above 5 micrograms per deciliter (µg/dL) globally. This paper reports on the potential risks of lead exposure from early life through later life. The articles used in this study were searched from databases such as Springer, Science Direct, Hindawi, MDPI, Google Scholar, PubMed and other academic databases. The levels of lead exposure in low income and middle-income countries (LMICs) and high-income countries (HICs) were reported, with the former being more affected. The intake of certain nutrients could play an essential role in reducing (e.g., calcium and iron) or increasing (e.g., high fat foods) lead absorption in children. Elevated blood lead levels may disturb the cells’ biological metabolism by replacing beneficial ions in the body such as calcium, magnesium, iron and sodium. Once these ions are replaced by lead, they can lead to brain disorders, resulting in reduced IQ, learning difficulties, reduced attention span and some behavioral problems. Exposure to lead at an early age may lead to the development of more critical problems later in life. This is because exposure to this metal can be harmful even at low exposure levels and may have a lasting and irreversible effect on humans. Precautionary measures should be put in place to prevent future exposure. These will go a long way in safeguarding the health of everyone, most especially the young ones.

## 1. Introduction

Heavy metal pollution has been a serious problem all over the world for many years now. Among such heavy metals is lead which is known to have come from a Latin word “plumbum” meaning “Lead” (*liquid silver*) and with a symbol of “Pb” [1,2]. Lead is naturally occurring and can also be present everywhere: in water, food, soil and in the atmosphere [3,4,5].

Exposure to this metal (Pb) has remained a serious environmental and public health problem worldwide due to its many uses over the past millennia. Its malleability and resistance to corrosion made it useful for many items, such as water pipes and vessels [6]. It also has numerous industrial and domestic uses, reflecting its unique chemical properties. However, lead exposure was not well characterized until the Industrial Revolution, when occupational exposure to lead became more common and began to affect workers and their families. The toxic effects of lead have been documented since ancient times, for example, in places such as the Roman Empire and Greece [6].

For example, the Romans ate food cooked in leaded vessels and drank water supplied through leaded pipes. Apart from these, they used lead as an artificial sweetener widely called (*sugar of lead* and scientifically known as *lead acetate*) as an addition to drinks, wine, liquors, cakes, sweets and all kinds of foods [7,8]. The entire Roman population became used to this leaded sweetener for years, which led to the buildup lead in their bodies. Most of them developed chronic health problems such as dementia, gout, anemia, cognitive impairment and impairment of neurodevelopment, especially in children [8,9]. Many of them even died from excessive and long-term consumption of this sweetener and other lead-contaminated foods and drinks [7,8]. Despite all these health problems and the death of many, consumption of these items continued as usual. In fact, some historians and scholars believed that the downfall of the famous Roman Empire then might have been due to lead poisoning [6,7,8].

Despite the knowledge of the health consequences that continue to result from lead exposure, especially in children, the use of lead in manufacturing various items such as plumbing, toys and household products has continued through most of the 20th century [6]. In addition, lead has also been used as an additive to paint due to its durability, resistance to moisture and the fact that it makes the paint dry quickly. Similarly, lead addition to gasoline was beneficial because it improved vehicles’ engines and prevented knocking [10,11,12].

### 1.1. Lead Exposure from Paint and Gasoline

During the past years, gasoline (also known as petrol) and paint used to be the major sources widely reported as the causes of lead exposure in many countries of the world. Reports about the continued danger resulting from the use of leaded gasoline and paint from different nations of the world became a cause of concern to international bodies. This led to a decision to phase out lead from these substances globally.

In 2002, an agreement was reached on the phase-out issue during the World Summit on Sustainable Development held in South Africa with representatives from every country [13]. This decision was also established in 2009 at the International Conference on Chemicals Management under the joint leadership of the UNEP and WHO to phase out lead from these substances [Strategic approach to International Chemicals management [14]]. Despite the decision to eliminate lead from paint and gasoline, the previous use of these substances and its aftermath effects continued to be a major problem [15,16,17,18]. For example, those houses painted with lead-based paint before the elimination still exist with people living in them [18]. Those already exposed to leaded gasoline when still in use have continued to grow with this metal in their bodies. In fact, it was lamented by Bellinger, 2008, about the United States that “… *it is a national disgrace that so many children continue to be exposed at levels known to be neurotoxic* …” [19] (p. 0691).

The ***United States*** for several decades experienced serious problems concerning lead exposure as a result of using lead-based gasoline and paint. This was because this country depended so much on leaded paint and gasoline, which started during the 1940s. The use of these substances led to a continuous release of high levels of lead into the environment, leaving many American populations with increased blood lead levels (BLLs), especially children who were more vulnerable to the harmful effects [20]. In fact, “… *over 170 million Americans alive today were exposed to high-lead levels in early childhood, several million of whom were exposed to five times the current reference level* …” [21], p. 1. Of the 318 million Americans, only 131 million had BLLs below 5 μg/dL [21].

Furthermore, it is reported that lead exposure was responsible for the loss of 824,097,690 IQ points in 2015 [21]. The issue of lead exposure did not only affect health, but it also affected the nation’s economy in so many ways. For example, the country spends about 50 USD billion annually on childhood lead exposure alone [22,23], not to mention other populations in the country. As a result of the severe health consequences, the government decided to eliminate lead from paint, in 1978 [24] and gasoline which commenced from the early 1970s through the 1990s [25]. This process led to a decrease in environmental lead release levels at that time [20]. Even though the elimination had taken place, the fact remains that many Americans had already been exposed to lead when lead-based paint and gasoline were still in use. For example, those who were children when these substances were still in use later developed many problems such as violent criminal behavior by the time they had grown into adults many years after [20].

***Australia*** is another country where there have been so many reports about lead exposure. This country is known to be “… *one of the world’s largest producers and exporters of Lead* …” [26], p. 1. Lead exposure used to be mainly from lead-based paint and gasoline, even though there were other sources of exposure like mining and smelting companies [26]. These, for several years, like in America, led to increased BLLs in the Australian populations and had severe long-term effects on the health of many people due to the toxic effects [16,17,26]. This led to the elimination of lead from paint and gasoline in 1996 and 2006, respectively [27]. Apart from the total removal of lead from paint and gasoline, one crucial measure put up by the Australian government to prevent lead exposure generally was by “… *educating people about potential Lead sources in their homes and environments*…” [22], p. 11.

***Germany*** has also been threatened with the issue of lead exposure for many years now. Exposure was majorly from leaded gasoline, which began around the late 1930s and led to widespread elevations in children’s BLLs that were discovered around the late 1960s [22]. Lead from paint was another major route of exposure but was not a big issue compared to gasoline. Although exposure to lead from paint was not seen as a severe issue, yet it was a cause of concern to the government. As a result of this, there was an agreement on the total removal of lead from paint in 2006 and 1998 for gasoline [22].

Apart from those two major sources described above, other sources of exposure were industry, vehicular exhaust fumes and children being exposed to parents already exposed to lead at work. Exposure to all these sources led to high levels of BLLs in children, especially those living in deprived areas and of lower socioeconomic status [22].

The story is no different for low-and medium-income countries concerning the dangers of using lead in paint and gasoline. Countries from these areas have suffered for many years from exposure to lead from paint and gasoline, which has also put the health of many in jeopardy; as a result, lead was removed entirely from these substances. For example, in ***South Africa***, the elimination of lead from gasoline and paint occurred in 2006 and 2009, respectively [28]. In ***Nigeria*** as well, lead was removed from gasoline in 2004 [29] and from paint in 2020 [30].

#### Trends of Lead Exposure Concentrations and BLL Monitoring

Several measures and mitigating steps were implemented by different countries in order to regulate the lead content of these substances. Some of these steps included constant clean-up of the past release of these pollutants, constant monitoring of BLLs to ensure they do not exceed recommended limits and educating people about the dangers of lead exposure. For example, Lermen et al. [31] reported that “… *Lead emission decreased in Germany continuously from the mid-80s due to different iterative mitigation steps* …” p. 4. These authors monitored the BLLs of some young adults in Germany between 1981 and 2019. The results of this longitudinal study revealed a substantial decrease (by 87%) in the BLLs of these young adults. Furthermore, the United States’ elimination of lead from these substances at that time led to a drastic drop in BLLs which improved the health of many Americans. For example, there was an increase in the cognitive function and intelligence quotient of children [21]. 

However, reports about the LMICs indicated that the BLLs of many people, especially children, continue to rise, and some have risen beyond 5 micrograms per deciliter (µg/dL) [32,33,34] even after the elimination of leaded paint and gasoline. This rise in lead levels in the LMICs is as a result of continuous exposure to lead from various other sources that are not being properly regulated [33]. As a result of this, many are still battling with many problems such as health and cognitive issues [32,33,34]. This could be because mitigating measures and steps (such as laws and regulations, constant monitoring of BLLs and other intervention programs) that could help reduce or prevent exposure might not be being properly implemented [33,34].

### 1.2. Common Sources of Lead Exposure

Apart from exposure to paint and gasoline, which have been widely reported in the literature, some other common and usual routes through which people are regularly exposed to lead and on a daily basis are food, soil, air and drinking water [3,4,5]. Exposure to lead from these various sources also remains a public health problem worldwide. One important point to remember is that, even though lead exposure affects the general public, children are more vulnerable since their organs are still growing and developing [32,35,36]. The United Nations Children Fund (UNICEF) reported that “… *Lead can be found throughout the environment in which children live—including in the air they breathe, the water they drink, the soil they walk/crawl on, the food they eat, the paint on the walls that they touch, and even in some of the toys they play with* …” [32], p. 26. A more critical case is fetuses exposed to lead right from the womb and at conception; the babies already had elevated BLLs and some health defects [37]. The intake of a diet rich in calcium during pregnancy could reduce or prevent the transfer of lead from mothers to the fetus [38].

Several studies have been conducted on the various common and usual sources (food, air, soil and water) of lead exposure over the years. Some of these studies reported concentration values [16,39], while some did not [8].

#### 1.2.1. Lead Exposure from Food

Food is one of the major pathways through which people (both young and old) can be exposed to lead. Ingestion or consumption of foods contaminated with lead could be the cause of many problems threatening human health today. For example, a study by Freeman et al. [40] reported that children’s contact with lead-contaminated dust from their environments introduced lead into their food (due to the children’s unwashed hands), and they later became exposed to lead through that route. It is also reported that occupational workers are exposed to lead from their workplaces and even “take home” the lead dust [41,42,43], which can be introduced into foods and later be ingested by other family members. For example, a study conducted in Pakistan among some lead-related occupational workers reported that the clothes of these workers contaminated the home environment, adversely affecting the health of the family members, especially the children [43].

Furthermore, regarding the ingestion of lead-contaminated foods, a report from the US indicated that many people are exposed to lead by consuming game meat because of the use of lead-based ammunition for hunting this game, which is a big problem in the US [22]. Lead from the ammunition is said to be introduced into this meat during hunting, and people eat the meat unknowingly. Even though the government has tried to stop the use of lead-based ammunition for hunting, the fact remains that many people still use them illegally, which has increased the BLLs of those who are lovers of, or are addicted to, game meat [22]. Other pathways through which lead can be introduced into food is from contaminated soil. For example, plants grown on lead-contaminated soil, especially those around mines and other related industries, will raise the BLLs of people that consume these food items. For example, a study conducted in Bangladesh of agricultural food products commonly consumed by Bangladeshi inhabitants, such as cereals (i.e., rice, wheat and maize), fruits and vegetables of various kinds. These food products contained very high concentrations of lead, which were beyond permissible limits. This was because these foods were grown on soils contaminated with lead through industrial operations, vehicular emissions and synthetic fertilizers. The authors further reported a risk of cancers of various forms [44].

On the other hand, apart from food grown on contaminated soils, studies have also reported that food items can be contaminated by dust containing lead from the roads, thereby causing health problems in individuals who ingest such foods. For example, Makhoka et al. [45] from Kenya conducted a study on the contamination of fruit and vegetables by road dust containing lead. These food items were displayed in the open air and retail shops. Samples of some of these food items (as seen in Table 1) were taken and subjected to laboratory analysis in order to determine lead concentrations.

The results of the analysis showed that lead concentrations in all these food items were far above the maximum recommended limits which may lead to elevated BLLs in the exposed individuals. It was advised that, to prevent future contamination, foods in the open air should be covered with thin-sheet plastics [45]. It was further reported that foods sold in the market or near significant roads could also be contaminated with lead from dust blown from the soil, air, or exhaust fumes from traffic.

One other area through which people can be exposed to lead through food is the consumption of aquatic foods contaminated with lead. Wastewater or effluent discharged into water systems may also contain a high amount of lead. When this gets into any water systems (such as streams and rivers), it tends to contaminate the water life and, thus, the fish and when they are consumed by human beings, they can jeopardize their health. A group of researchers conducted a study around a lake in Kenya. Samples of some of the fish were taken and subjected to laboratory analysis. The results revealed that all the tested fishes had an amount of lead above the recommended limit of 0.2 (µg/g) [45]. The authors went further to report that the health of the people in this area was at risk. Serious measures must be implemented to ensure people consume foods safe from environmental pollutants. Kumar et al. [46], in their study, also reported that wind sometimes can blow soil that has already been contaminated into homes thereby contaminating food items and even drinks.

#### 1.2.2. Lead Exposure from Soil

Many soils in different parts of the world are contaminated with heavy metals such as lead, which have deteriorated those soils. Some of the causes of soil contamination today could include proximity or being adjacent to smelting industries, lead mines, lead-acid batteries, automobile traffic and wastewater from industries. Food crops grown on any of these soils may contain high amounts of lead which may raise the BLLs of the people that consume these food items, thereby putting their health at risk [45,46], for example, from plants grown on lead-contaminated soil, especially those around mines and other related industries. For example, a study was conducted in some mining communities in Nigeria. Here, samples of some agricultural food items such as vegetables (lettuce, tomatoes, red peppers, potatoes, cabbages, onions, carrots and green beans) samples were collected and analyzed. The results of this study revealed that the amount of lead found in these vegetable samples were far above the recommended limits [47].

Apart from food items, samples of soils were collected from these mining communities and the results of the analysis also showed that the levels of lead in the soils were largely above the recommended guidelines [47]. The authors went further to assert that the health of the inhabitants living in these communities was at risk. Apart from the people living around the mines, those working in the mines may be even more affected and their family members could also be affected through them. It was reported in the study of Steffan et al. [48] that those in professions that have to do with the soil, such as mine workers, may be at risk. These authors went further to emphasize these mine workers could also carry this contaminated soil dust into their homes and that the family members, especially children that walk around the house barefooted, could be exposed to lead [48].

Farmers are also reported to be at a greater risk of lead exposure since they are permanently working with soil. In a situation where the soil is contaminated with lead, especially those farms close to mines or other related firms, they can easily be exposed to this metal [48].

It was also indicated in a study by UNICEF [32] that lead exposure from soil is a major cause of human health problems all over the world especially in the LMICs. For example, samples of soil were taken from around some lead battery recycling plants from seven African countries which included Ghana, Kenya Nigeria, and Tunisia. The results of the analysis revealed that “… *Lead levels ranged from <40 ppm to 140,000 ppm, with 81 per cent of soil samples having lead levels greater than 80 ppm* …” [32], p. 29.

It should also be noted that dust from lead-contaminated soil can be blown by the wind into homes and school classrooms where children learn and also affect the general environment.

The Hawaii Department of Education, in the USA, found that many soils are contaminated with lead in specific communities where children play and it may be unintentionally ingested. This prompted the Department to conduct a study where soil samples around 23 schools on the eastern side of Hawaii Island were collected and analyzed. The results showed that almost 60% of the schools had areas with elevated soil lead levels as a result of this contamination [49]. The implication is that continuous exposure to these contaminated soils could jeopardize these children’s health.

Laidlaw and Taylor [17] conducted a study in Australia of children exposed through lead-contaminated soil from mines. The blood lead levels among the children were reported to be around 0.707 mg/m^3^, and health problems such as low educational scores and developmental vulnerabilities were reported.

Another study was conducted in New Orleans among school children who lived in a community where the soils were mixed with lead and some other metals. This study determined the effects of lead on the academic performance of grade 4 children. The results of the study revealed that those children exposed to lead underperformed in the school standardized tests [50].

#### 1.2.3. Lead Exposure from Air

Air is another widespread route through which people become exposed to lead. Pollutants, including lead, are generally emitted from industrial activities such as mining and the combustion of fossil fuels. People close to these polluting industries are at a higher risk of exposure. Apart from this, soil dust around these industries may be blown by wind and people around or even far away from these industries may be exposed to lead by inhalation, thereby putting their health at risk [48].

Similarly, coal dust from coal mines may contain some amount of lead which can be blown by strong winds into the air, especially from storage yards or during transportation. Inhalation of this coal dust can result in chronic health problems to those exposed [51].

Rasnick et al. [52] studied the health effects of lead exposure on 263 children. They found out that exposure to lead from the air was responsible for different health and behavioral problems such as increased anxiety, aggression and attention problems observed among these children.

A similar study was conducted in China of 379 children that lived in seven (polluted) villages around lead mines and processing plants and another control group of 61 children from another village. This was achieved by determining the BLLs of the children and taking samples from their environments. The results of the study revealed that lead concentrations were higher in the lead-polluted villages compared to the non-lead-polluted ones. This indicated that the children in those lead-polluted villages had elevated BLLs compared to those from the non-lead-polluted village (87%, 16.4 lg/dL vs. 20%, 7.1 lg/dL). The authors concluded that “… *the lead industry caused serious environmental pollution that led to high BLLs in children living nearby* …” [53], p. 834. This further resulted in some adverse effects on many children’s health, including anemia, behavioral problems, such as hyperkinetic syndromes or attention deficit disorder, and digestive disorders such as stomachache, abdominal distension, or partiality to a particular kind of food.

#### 1.2.4. Lead Exposure from Drinking Water

Some studies have also reported on lead exposure through drinking water from different parts of the world. One example was the case of the Roman Empire [7], also, in the USA [54,55] and the UK [56]. In some poor cities of wealthy countries, such as in Flint Michigan, USA, it was reported that the drinking water in this community contained very high levels of lead which has been a major public health problem for several years, most especially among children [55,57]. For example, it was reported that 5 per cent of some children that were tested in Flint, Michigan, had elevated blood lead levels which were above 5 µg/dL and twice the recommended standard [32]. For example, it was lamented that “… *the water in one home tested contained 13,200 ppb of lead, more than two and half times the level of lead that could be classified as “hazardous waste” by the US EPA* …” [32], p. 42. Several other studies have also reported cases of lead exposure from other LMICs.

Kordas et al. [33] conducted a study in some Asian countries such as India, Indonesia and Philippines on lead exposure through drinking water from hazardous waste sites. It was reported that lead levels in the water samples were above recommended standards. The authors went further to report the predicted BLLs of 0–9-year-old children which ranged from 3 to ~60 µg/dL [33]. Similarly, samples of drinking water were taken from different water sources (25 wells, 15 boreholes, 7 taps and 3 streams/rivers) in five communities in Southeastern Nigeria and subjected to standard analysis. The result of the analysis revealed that the concentrations of lead in all the water sources tested were far beyond the recommended standard of the US EPA (15 lg/L (0.015 mg/L) [58].

In South Africa, Olufemi et al. [59] assessed the levels of lead in drinking water samples from two secondary school taps and a river from a South African coal mining community (as seen in Table 2). The analysis revealed that the lead levels for all three locations were at the South African Standard for drinking water (SANS) threshold. This shows that lead was present in the drinking water that the children and community members drank.

The authors of this study went further to aver that even though it might look as if the levels of lead reported from these three locations may not be that high as compared to levels reported in other studies, the issue being raised here was the presence of lead which at any level could be harmful. It was observed that the children in this community had been drinking this water from childhood, which may have a long-term effect on their health. Some of these children may sometimes fall ill and be absent from school, which may be due to the prolonged intake of water. To confirm this, Amadi et al. [60] reported that long-term exposure to lead could result in chronic health problems.

The study of Ahangar et al. [61] focused on lead exposure from multiple sources. This study was conducted on some children living around industrial areas in Tehran. They were referred from two pediatric gastrointestinal clinics with different ailments, such as abdominal pain and constipation. Their blood samples revealed that these children were exposed to lead at concentrations far higher than the Environmental Protection Agency (EPA) guidelines. Furthermore, the authors reported that these children were exposed to lead from various routes such as scratched wall paint, house floor dust, windowsill dust, tap water, and consumption of spices [61].

Other than the above-mentioned sources, there are some other known and unknown sources of lead exposure which have also been a major problem, especially in the deprived communities and poor areas of some wealthy countries. These other sources may include the manufacturing of local household products such as lead-glazed clay pottery [62], welding, metal smelting, artisanal manufacturing of wares, such as jewelry and decorative items, transportation of products or substances that may already be contaminated with lead [33]. Industrial lead emissions such as from mining, the ingestion of an Ayurvedic local medicine (which is reported to contain a very high content of lead), consumer and household products containing lead, lead-acid battery recycling, subsistence fishing, incineration, electronic waste, cultural or traditional practices (geophagia) [39], lead melting for making fishing sinkers [39]. For example, the blood samples of 160 school children in two fishing villages in the coastline areas of South Africa were taken in order to determine the concentrations of lead. The results of the analysis showed that these school children had elevated BLLs as a result of continuous exposure to lead from these activities. These authors went further to report that “… *Lead is melted by adults, often in the presence of children, and sometimes children themselves melt lead, for use in fishing activities or to play with*…” [39], p. 183. Mouthing, as well as unintentional ingestion of lead sinkers by children was reported by these authors [39]. A similar case was from Cartagena, Colombia, where it was reported that the majority of Colombian populations with the highest BLLs lived in the fishing communities where the use of lead sinkers for making fishing nets is the major business [33]. It is reported that e-waste recycling is a major source of lead exposure in African and Asian countries and that most of these activities are performed within the home environments which can negatively impact the health of many [33]. In Mexico, the widespread use of handmade lead-glazed clay pottery for cooking, storing and serving food is a major public health concern [62]. These activities for many years have greatly contributed to high levels of lead in this country and have put the health of many Mexican populations at risk [62]. One major problem with these other sources is that some of them are difficult to regulate, unlike paint and gasoline. This may be due to many factors, such as “… *poverty, a large informal sector, competing for public health challenges, low levels of awareness of Lead hazards and weak capacity to enforce legislation* …” [39], p. 1.

## 2. General Health Effects of Lead

Lead, a toxic metal, has been studied for many years by scholars worldwide who have consistently reported that exposure to this metal either by inhalation, ingestion or skin contact can result in various health problems in humans. Even though it affects both adults and children, it is to be noted that children are more vulnerable to the adverse effects due to their developing organs and nervous system. Several studies have reported that there is no safe blood lead level, and even small exposures can result in a reduced intelligence quotient (IQ), the inability to pay attention, poor academic performance and other health problems in children [23,32].

### 2.1. Effects on Adults

Millions of adults worldwide suffer from many health problems due to environmental pollutants such as lead. This can be confirmed in the studies of [63,64], where it was indicated that many of the health problems some adults experience today could be as a result of being exposed to lead while they were still children. Rees and Richard [32] indicated that “… *over 900,000 premature deaths per year are attributed to lead exposure* …” p. 1. Studies conducted worldwide (as seen in Table 3) have reported some health outcomes of lead exposure in adults. For example, a study was conducted in New Zealand on some students exposed to lead in childhood. These were followed up till they grew up to be adults. The results of the study revealed that “… *Childhood lead exposure may have long-term consequences for adult mental health and personality* …” [64], p. 418.

It is widely reported that most of the adults affected by lead are occupational workers (i.e., those who work in industries such as mines, smelting companies, and lead acid batteries) [32,65] compared to other members of the same populations. In fact, according to the U.S. Occupational Safety and Health Administration (OSHA), around 804,000 industry workers and 838,000 construction workers are exposed to lead through these routes [66]. Occupational workers are exposed to this toxic metal daily and even “take home” the lead dust [41,43] which can be introduced into foods, and drinks and transferred to other family members. For example, a study conducted in Pakistan of some lead-related occupational workers reported that the clothes of these workers contaminated the home environment, adversely affecting the health of the family members, especially the children [43].

Adults exposed to lead are reported to likely suffer from various health problems such as high blood pressure, kidney and brain damage, cognitive decline, cardiovascular diseases, irritability, headaches, and hallucination, which can later progress into convulsions, infertility in males, damage to the nervous system, miscarriage in pregnant women, paralysis, and even death [8,67,68,69].

**Table 3 ijerph-19-16006-t003:** Summary of epidemiological and scientific studies on health effects of lead exposure on adults from different countries.

Authors	Location	Population	Study	Exposure	Health Outcomes
Obeng-Gyasi [70]	USA	Adults	Cross-sectional	-	Cardiovascular diseases
Reuben et al. [64]	New Zealand	Adults	Longitudinal	-	Long-term psychiatric and behavioral consequences
Steeland et al. [71]	USA, Finland, UK	Adults	Cohort	Industries such as battery, smelting and railroad	Lung cancer, stroke, chronic obstructive pulmonary disease, stroke and heart disease
Wani et al. [8]	India	Adults	A review	-	Anemia, brain and kidney damage, miscarriage, infertility
Lamphear et al. [72]	USA	Adults	Longitudinal	-	Cardiovascular diseases, ischemic heart disease and mortality from these diseases were reported
Ahmed et al. [65]	Ethiopia	Adults	Cross-sectional	Lead acid battery industry	Visual problems, asthma, gastrointestinal and kidney problems reported

### 2.2. Effects on Children

Lead exposure is a dreadful public health problem that affect many people all over the world and majority of those affected are children [32]. To substantiate this fact, it was indicated in a UNICEF report that “… *around 1 in 3 children—up to approximately 800 million globally—have blood lead levels at or above 5 micrograms per deciliter (µg/dL)* …” [32], p.1.

Numerous studies (as seen in Table 4) have reported that lead exposure from any of these sources could result in various illnesses and even premature death among children [7,42]. Furthermore, some studies have specifically indicated that children whose fathers work in factories such as mines and smelting companies are exposed to lead around their homes through their fathers’ contaminated clothes, which adversely affects their health in various ways [8,32,43].

One important thing to note about lead is that the higher the level of exposure, the greater the health consequences [8,73]. Some health consequences of lead exposure among children have been reported [42,74]. These include anemia, nervous system effects, abdominal pain, colic, vomiting, hearing loss, dental problems, convulsion, and other health problems. For example, in South Korea, it was reported that children with elevated blood lead levels were diagnosed with oral health problems such as plaque deposition and gingival diseases [74]. Although lead exposure affects almost every part and organ of the body, the nervous system is the most targeted [8,75]. Many studies have reported permanent brain or nervous system damage due to prolonged exposure to lead [32,75,76].

**Table 4 ijerph-19-16006-t004:** Summary of epidemiological and scientific studies on health effects of lead exposure on children from different countries.

Authors	Location	Population	Study	Exposure	Health Outcomes
Tort et al., [74]	South Korea	Children and adolescents	Cross-sectional	Through ingestion of food and water	Plaque deposition and gingival diseases
Khan et al., [43]	Pakistan	Children/adults	Cross-sectional	Lead-contaminated clothes of fathers (who are occupational workers)	Hematopoietic, renal, and hepatic functions
Moore et al. [7]	Roman Empire	Children	Historical/cross-sectional	Different sources such as water, food and sweeteners	Skeletal evidence of metabolic diseases in both groups studied was reported. A high rate of mortality, especially among children, was also reported
Mathee et al. [39]	South Africa	Children	Cross-sectional/analytical	Lead smelting	Children’s health at risk
Laidlaw and Taylor [17]	Australia	Children	A review	Lead-contaminated soil	Health problems such as kidney damage, dental caries, ADHD, and IQ decline were reported
Dong et al. [16]	Australia	Children	Experimental	Aerosols, dust and soils from mines	Low educational scores and developmental, vulnerabilities reported
Lin et al. [53]	China	Children	Experimental/longitudinal	Airborne through lead mines and processing plants	Anemia, hyperkinetic syndromes or attention, deficit disorder; stomachache, abdominal distension
Rasnick et al. [52]	USA	Children	A prospective birth cohort study	Air	Low-level lead exposure from the air in childhood resulted in poor behavioral problems such as aggression at age 12. Other problems reported are increased anxiety and attention problems
Zahran et al. [50]	New Orleans, USA	Children	Cross-sectional/case control	Soil	Poor school performance
Olufemi et al. [59]	South Africa	Children	Analytical	Drinking water	Potential health risks such as respiratory infections stomachache and diahorrea reported
Sharma et al. [77]	India	School children	Cross-sectional	Traffic (vehicular emissions)	An elevated BLL was reported in these children compared to their western counterparts. Risks of behavioral problems, fatigue, stomachache, poor academic performance reported
AbuShady et al. [78]	Egypt	Children	Cros-sectional	Different sources such as air, water, dust	Abnormal behavior, pallor
Roberts et al. [79]	United Kingdom	Children	Case study	Not reported	Neurobehavioral problems, pica, developmental delays reported. Children from deprived areas are at a higher risk
Freeman et al. [80]	USA	Children	Survey	Lead-contaminated homes (i.e., dust and chips from old paint	
Mbewe et al. [81]	Zambia	Children	Cross-sectional	Abandoned mine	Of the studied children, 99.4% were at risk of chronic acute health problems
Pradhan et al. [82]	India	Children	Cross-sectional	Drinking water	Anemia due to iron deficiency reported among studied children
Rubio-Andrade et al. [83]	Mexico	Children who were tested for lead at 0, 6, 12 and 60 months	Longitudinal	From metallurgy smelting companies	The risk of anemia was reported among children
Hashim et al. [84]	Malaysia	School children	Comparative study	Air, foods, water, soil and paints,	Respiratory problems

Looking at the above accounts about lead and reports from different countries, one realizes that there are different and various health consequences that can arise from being exposed to this metal, even at the lowest levels of exposure. This is confirmed in studies where it is reported that lead exposure can be harmful even at the lowest level [85,86,87]. One important thing to bear in mind is that even though lead exposure is an issue throughout the world and remains a major public health problem, the fact remains that those in the low- and middle-income countries (LMICs) and the poor areas of the high-income countries (HICs) are more affected [32].

## 3. Lead Exposure: Low- and Middle-Income Countries versus High-Income Countries

Lead exposure has continued to be a severe problem worldwide over the past years due to human activities. This has led to the deterioration of water, food, soil and air and has further affected the health of many, especially the vulnerable population groups [33]. For example, individuals exposed to lead in childhood were reported to manifest different problems such as cognitive deficit with downward social mobility by the time they were adults. Apart from its effects on humans, lead exposure has also greatly affected the economy of various nations in several ways [22,23,32,33].

There is no exception when it comes to lead exposure. Even though exposure to this metal is a general phenomenon which cannot be underestimated, it is to be noted that people from the LMICs are more affected [32,33]. A report by the United Nations indicated that “… *92% of all deaths attributable to Lead exposures occur in Low- and Middle-Income Countries …*” [32], p. 8. This means that lead exposure rates are higher in LMICs than in HICs [34]. Industrial activities such as mining and smelting are reasons for this high exposure rate. Furthermore, the production and manufacturing of essential and highly demanded substances such as batteries, local guns and bullets, jewelry, lead-glazed pottery for cooking, sinkers for fishing nets and some substances that require welding [33]. Other than these, people from many communities in the LMICs continue to engage in some practices and acts that expose people to lead. As reported by [33], some of these activities may include the continued burning of toxic substances such as tires, e-waste, discarded rubbers, medical waste, and paper products. People who live close to these burning sites may be affected by regular inhalation of their fumes. Apart from exposure through industrial activities, which may require government and international intervention, one of the major reasons why the problem of lead exposure has continued in these areas may be because some of those other exposure sources seem to be well established and somehow permanent, which may also require behavioral change.

Children are the most significant population or group of people being threatened by lead exposure due to their developing organs and behavioral patterns [32]. To support this claim, the United Nations Children’s Fund (UNICEF) reported that many children worldwide are exposed to lead, which affects their health in many ways [32].

Most children affected by lead live in poor areas of the world [32,33,88]. It was reported by [34] that, despite the elimination of leaded petrol, “*Many children have a blood lead level exceeding 5 µg/dL in LMICs* …” p. e145.

African and Asian populations are in the categories that are seriously affected by lead exposure, which has put the health of the people, especially children, in jeopardy [32,33,88].

For example, Kabwe and some surrounding communities in Zambia are areas that for many years have suffered due to pollution from lead smelting companies and an abandoned lead mine which was closed over three decades ago. Although the lead mine is no longer in use, there has been continued exposure, which has seriously affected the soil, water and the general ecosystem. Several studies conducted in the area have reported that lead exposure has put the health of many Zambians at risk, especially children with elevated BLLs beyond recommended limits [89,90,91,92]. Similarly, in Nigeria, studies have reported that lead exposure from different sources threatens the health of many children, especially in the local areas [93].

For example, Kaufman et al. [93] conducted a study in some villages located around the gold ore processing area in Zamfara State, Nigeria. They reported an outbreak of acute lead poisoning from the gold ore processing, which “*killed more than 400 children ≤5 years of age in the first half of 2010 and has left more than 2000 children with permanent disabilities* …” p. 2.

Even though lead exposure may seem to have been reduced in high-income countries (HICs) due to severe measures that have been put in place to control it, the fact remains that there are still some poor or deprived communities in the HICs which are seriously affected by lead exposure from various sources [79].

For example, Cincinnati, one of the poor communities in the USA, has been known for many years now to be a hot spot for lead exposure, with several children’s health being endangered due to continuous exposure to this metal [94]. Over the years, exposure to lead for the people of these communities would already have been a part of their daily and normal life, with many of them not even realizing the health effects.

Furthermore, Flint, in Michigan, USA, has for several years been threatened with the issue of lead exposure from drinking water which has affected the health of many in this deprived city, most especially the children [55,57].

Pradhan et al. [82] conducted a study of Indian children exposed to lead. The prevalence of anemia was reported in some of the studied children.

A similar study was conducted in one local area of India of school-age children constantly exposed to lead. The authors of this study reported elevated BLLs, which were far above the recommended standard in these children compared to other children [52].

The United Kingdom, a high-income country, is being threatened by lead exposure (both in urban and rural areas). Levels of exposure are lower in urban areas than in rural areas due to legislation and measures put in place by the Government. However, those in deprived or poor areas are more affected by lead exposure compared to those in urban cities. For example, some lead-emitting companies are situated mainly in these rural areas. One can see from a study conducted in England by [79] where it was reported that “*children from deprived areas may be at higher risk of Lead exposure* …” (p. 548).

A study was conducted in the rural part of Northern Mexico of children living around a metallurgy smelting company. These children were tested for lead at 0, 6, 12 and 60 months. The BLL concentrations obtained for each of these months (0, 6, 12, and 60 months) were: 10.12 μg/dL, 8.75 μg/dL, 8.4 μg/dL and 4.4 μg/dL, respectively. Risk of anemia was observed in these children [83].

China, another high-income country, is known to be the largest producer of lead in the world. As a result, many Chinese populations (both urban and rural dwellers) are affected by lead exposure, resulting in various health problems [95,96]. For example, Zhang et al. [95] conducted a study of lead exposure among Chinese children, and they found that the BLLs of children living in rural areas were far higher than those in urban areas. Another critical issue was that the significant and highest sources of exposure for children in rural areas were contaminated drinking water, diet [97] and soil [98].

### Challenges of Reducing Lead Exposure

It has been established that when it comes to lead exposure, LMICs and poor communities in the HICs or wealthier nations are more affected. Even though efforts to reduce lead exposure have always been in place and on a continuance basis, the fact remains that various challenges are being encountered in making this a reality. Some of these challenges are highlighted. For example, the study of [33] reported about insufficient scientific data and decline in research publications in the area of lead exposure for some years now; a lack of biomonitoring programs; a lack of knowledge about the danger of lead exposure on the part of businesses owners. [15] reported about South Africa and other LMICs, some challenges that prevents lead exposure in places such as cottage industries, the informal sector and from traditional practices. These challenges may include limited or lack of resources to enforce occupational and safety legislation. This author pointed out further that these businesses (the informal sector and cottage industries) and traditional practices are not registered or regulated, and they sometimes operate secretly and illegally [15].

One of the best ways to reduce exposure in these poor communities is to totally ban or reduce the number of those manufacturing or small-scale businesses. However, because these businesses have long been operating, stopping them may be a challenge since this is what many of them have been depending on for a living. Other challenges may include people’s inability to stop regular practices such as the burning of waste.

The marginalized or poorest of the poor populations are likely to live in proximity to lead exposure sites due to illiteracy and low levels or lack of awareness of the danger. One way to stop them from continuous exposure is for the government to think of relocating them far away from these exposure sites.

Failure to reduce lead exposure will continue to lead to increased lead absorption, especially among vulnerable population groups such as children. One of those suggested precautionary measures to prevent the process of lead absorption is through proper nutrition [99].

## 4. Lead Absorption and Nutrition

Nutrition or diet plays a vital role in reducing or increasing lead absorption in children and has been well documented in the literature. Certain foods with high fats, low minerals, low fibre, low iron, and excessive or low proteins are reported to increase the absorption of lead in the body [99]. Continuous intake of these foods will encourage or increase lead absorption among children. For example, a study in New Jersey, USA, revealed that children who consumed fatty foods such as hamburgers, doughnuts, peanut butter and jelly sandwiches, and cold cuts had elevated BLLs [80]. On the other hand, foods rich in iron, calcium, lactose, zinc, and various vitamins are very helpful in fighting lead absorption [100,101]. For example, a study conducted on lead concentrations among some Uruguayan children revealed that children that consumed more calcium, milk, dairy, and yoghurt were reported to have low BLLs [102]. Similarly, the study of Freeman, et al. [80] indicated that children who were fed with food rich in vitamins, raw vegetables, and yoghurt had lower BLLs.

On the other hand, it was reported that some children were continuously exposed to lead through a metal transporting company and that iron deficiency was responsible for elevated BLLs in those children [102]. When it comes to lead absorption, one important fact we need to know is that children from the LMICs are likely to be disadvantaged compared to children from high-income areas [79]. The reasons may be due to poverty, ignorance or lack of knowledge about the issues of lead exposure, especially for parents.

The parents of children from the LMICs or poor areas of HICs feed their children just any type of food because many of them may be illiterate and not have any understanding or knowledge about these matters. Even if they are informed about these issues, they still might not be able to afford to purchase the correct food nutrients needed to fight lead absorption compared to parents of children from the HICs. In these HICs, most parents are literate, affluent, and have adequate information about lead-related issues and how to fight or prevent them, although lead exposure may be higher in the LMICs or poor areas of the HICs, as reported in many studies [79,81]. If children are fed with foods with those nutrients that can help reduce the absorption of lead, there will be little or no concentration of lead in their blood [102]. Therefore, parents and teachers at different schools must be adequately educated about lead exposure. Not only that, but they also need to be taught about how the intake of certain nutrients or a healthy diet can assist in fighting lead absorption. Taking all these necessary preventive measures will go a long way in protecting the health of these children.

## 5. Possible Health Consequences of Lead Exposure in Adulthood

In a situation whereby children are not given the necessary help to combat lead absorption, there will continue to be an accumulation of this metal in their blood through various sources as they grow. As the level of absorption increases, there is a tendency for the health effects also to continue to increase [8,73]. As they continue to grow, by the time they reach adulthood or old age, there is a likelihood of developing more critical problems such as kidney damage [32], neurodegenerative [63] and cardiovascular diseases [70,103,104], These are major public health problems and leading causes of death in many nations today, most especially in the United States. For example, in a study conducted by [63], an association between lead exposure in childhood and increased risk of neurodegenerative diseases (such as Alzheimer’s and Parkinson’s diseases) in adulthood was reported. This author further emphasized that most people diagnosed with neurodegenerative diseases at old age were people exposed to lead at early ages [105].

With regard to cardiovascular diseases, a study [70] conducted in the USA reported an association between lead exposure and the likelihood of cardiovascular diseases among young and middle-aged adults. The author reported higher BLLs for middle-aged adults than younger adults due to prolonged exposure, “… *which has put them at risk of diseases such as cardiovascular disease* …” [70], p. 5. Similarly, Xu et al. [104] reported a relationship between lead exposure and cardiovascular diseases in some US adolescents. In their study, Yan et al. [105] also explained that cardiovascular disease from lead exposure is not yet an issue in low-income countries compared to high-income countries. However, these authors specifically warned that urgent measures need be taken to combat lead exposure in low-income countries to prevent a high rate of cardiovascular and other related chronic diseases in the future [105].

Another major problem that could arise from prolonged exposure to lead, as reported in some studies from different parts of the world (Australia: [26]; USA: [106,107,108,109] South Africa; [110], is violent or criminal behavior. These studies reported on people exposed to lead at younger ages and how they developed violent or criminal behavior later in adulthood. For example, Nevin [111] conducted a longitudinal study on the relationship between lead exposure from gasoline and violent crime in the United States. The results of the study revealed “… *that the per capita consumption of leaded gasoline is strongly associated with all types of violent crime* …” [111] p. 215. Similarly, the study by [108] found a positive correlation between lead exposure and crime rates in the United States. Some children manifest minor behaviors early, such as aggression and delinquency, and grow up with increased BLLs in their blood. These behaviors that seem to be minor may likely graduate into more harmful acts, such as criminal behavior, later in life. This is confirmed by [109,112], where an association between lead exposure and the manifestation of aggressive and delinquent behaviors were reported among those children.

All these claims can be supported by the lead–crime hypothesis, which states that (1) lead exposure at a young age leaves children with problems such as learning disabilities, ADHD, and impulse control problems; and (2) those problems cause them to commit crimes as adults—particularly violent crime [107,113,114,115].

It is to be noted that the potential futuristic effects of lead exposure mentioned above have mostly been reported in HICs, such as the United States, compared to the LMICs. Despite all the various measures put in place to combat lead exposure in these HICs, these issues remain public health problems in these countries. One important thing to remember is that lead exposure is higher in the LMICs compared to the HICs, and the LMIC populations are more affected by the consequences. It is therefore very important that urgent actions are taken to prevent people from LMICs, especially children, from being exposed to lead from any source. This is because exposure to this toxic metal has damaging and lasting effects on human health. Preventing lead exposure in early childhood will go a long way in preventing problems that may occur in adulthood or later in life. This is confirmed by [63] who asserted that “… *Millions of Americans now entering midlife and old age were exposed to high levels of lead, a neurotoxin, as children* …” p. 1.

## 6. Lead Exposure: Implications for Public Health Education

Although lead exposure has been a serious public and environmental problem for several decades now, all over the world, it is unfortunate to note that many people all over the world have little or no information about this metal or its sources and even less of its’ resultant effects or dangers [116,117]. Even those who have already been exposed and manifesting some health effects lack awareness about these issues.

For example, a study conducted in South Africa among some pregnant women reported a low level of awareness among these mothers [117]. Similarly, *Adebamowo* et al. [116] conducted a study in Nigeria on the knowledge, attitudes and practices related to lead exposure. The result of the study revealed that “… *a limited awareness of the sources of lead exposure in the domestic environment and participants had little knowledge of the health effects of chronic low dose lead exposure* …” p. 1. Looking at reports from other studies conducted around the world, it can be deduced that there is a general lack of awareness and knowledge about issues related to lead exposure [118]. Therefore, the government of every nation must formulate a policy that will ensure that every citizen is educated correctly about these matters, especially the younger generations, through all possible means and avenues. These could include school lessons, intervention programs, seminars and workshops, and media such as television, radio, internet and newspapers. This will go a long way in helping everyone to be informed about lead exposure sources and dangers and how to take the necessary steps and actions to prevent it collectively. A popular adage says, “prevention is better than cure”.

## 7. Methods of Selection of Articles

The relevant literature used in this study was selected through a careful and thorough search and from various databases such as Springer, Science Direct, Hindawi, MDPI, Google Scholar, PubMed and other academic databases. The search was carried out by using string/keywords, which included: an overview of top priority heavy metals, such as mercury (Hg), cadmium (Cd), arsenic (As), chromium (Cr), zinc (Zn) and lead (Pb). After reading through some of these different heavy metals, even though they all pose a significant danger to human health, it was discovered that lead is the most dangerous because the effects are irreversible and cannot be corrected once exposed. So, the authors opted for lead which later became the focus of this study. Various other keywords searched were “Nature of lead”, “Origin and history of lead (with a focus on the Roman Empire)”, “Lead exposure generally”, “Major sources of lead exposure (focusing on gasoline and paint from different countries, both wealthy and low/medium income countries).” Other common sources of lead exposure which included “Lead exposure from air”, “Lead exposure from drinking water”, “Lead exposure from the soil” and “Lead exposure from food”, “Health effects of lead exposure on children”, “Health effects of lead exposure on adults”, “Lead absorption in children (compare LMICs and HICs)”, “Role of nutrition in combating lead absorption”, “Lead exposure from both HICs and LMICS”, “Lead exposure and preventive measures”, “Possible later life effects of lead exposure”, “Lead exposure and cardiovascular diseases from both HICs and LMICs”, “Lead exposure and violent criminal behavior”, “Lead–crime hypothesis”, “Lead exposure awareness”.

Most articles searched and cited in this study were from 2000 and 2022, except for some articles published before 2000, which were selected based on their relevance even though they might seem old, i.e., [6,25,88,99,100]. Hundreds of peer-reviewed articles were initially searched and carefully studied for this study, but only those most relevant to this study were selected, while others were put aside for future use. Many of the journal articles used in this study are highly cited articles with high impact factors. Some journal articles with low or moderate impact factors were also included, based on relevance.

## 8. Conclusions

This review reports on existing literature about the potential health risks of lead exposure from early life through to later life.

Firstly, a brief description of lead and the various sources was made. This was followed by the general health effects, especially on children known to be the most vulnerable. These general health effects may begin to manifest right from birth through the childhood stage. As they grow with this metal in their bodies, they could develop more severe problems such as cardiovascular diseases and violent or criminal behavior later in life. There is no safe level for lead; even small exposures can lead to severe problems in the exposed individuals. Exposure to lead from any sources should be avoided as much as possible. This is because the effects are reported to be permanent and irreversible and may not be corrected once exposed. Precautionary measures should be taken to prevent exposure due to its lasting and irreversible effects. If lead exposure has detrimental effects on animal [67] and plant health [119], as reported in the literature, it will be much more detrimental to human health.

## 9. Pre-Emptive Measures

This paper reports on the potential health risks of lead exposure from early life through adulthood. Based on reports from several studies, individuals exposed to lead from the womb or at a young age will continue to manifest other health and behavioral problems as they grow up. If care is not taken, they tend to develop into more serious problems by the time they reach adulthood or old age. Based on this reality, preventive measures must be put in place to forestall such irreparable effects.
Reports from different nations have made it clear that even though there has been a phasing out of lead from gasoline and paint, which were the major sources of lead exposure in almost all the countries of the world during the past years, the fact remains that paint chips and dust from old houses are still a major source of exposure up till today [56,120]. HICs such as the USA are already coming up with strategies for removing old paint and replacing it with unleaded paint [22]. Therefore, the governments of the LMICs, with the assistance of international bodies, should consider developing a policy that will aim at following the same process of replacement ongoing in the HICs.There is a possibility that the unleaded gasoline that replaced the old leaded gasoline might contain some toxic compounds [121] that unknowingly may already be a problem or become a major problem to public health. Early investigation of this may be necessary.Proper medical checkups should be put in place to monitor children’s blood lead levels and health status from time to time.Proper awareness or education [118] should be given to concerned stakeholders such as educational management, industry owners (such as smelting and mining), community leaders, parents, health care officials and government representatives about the danger of this metal and the various ways exposure can be prevented. This education is crucial because it will go a long way in helping them protect these minors from relevant sources of exposure early in life. Apart from these various stakeholders, children should also be educated about lead exposure, its sources and the dangers as they grow.Having proper education [118] about these issues will go a long way in changing negative behavior to positive and preventing future problems.In the case of areas where past use of leaded paint and gasoline has contaminated the soil, children must be kept away from such areas. In addition, soil remediation could be embarked on from time to time to prevent future exposure [33].Children from the LMICs should be given access to good healthcare, safe facilities and proper nutrition to fight lead absorption [122].Laws and policies should be enforced to keep the environments where children live and learn safe and free from lead exposure or other environmental contaminants [32].Many children born these days have already been exposed to lead right from the womb. To prevent this prevalence, it is very important that proper education be given to pregnant women to stay away from any sources of lead exposure as much as possible for the sake of the fetus in the womb. In addition to this, education should be given to mothers during pregnancy about the intake of foods rich in nutrients such as iron and calcium that can prevent the transfer of toxins to the fetus [37,38].These are also applicable to lactating women [123].Global and international actions and measures that are put in place must be adhered to by every country concerning lead exposure from various sources and other related environmental pollutants to ensure that this planet is safe for all to live in [32].Research studies relating to the long-term and later life effects of lead exposure should be strengthened; for example, longitudinal studies are advised in order to reach concrete conclusions [33].There may still be other sources of lead exposure which may not even have been identified. These should be investigated.Extensive scientific data is needed for generalization [33].

## Figures and Tables

**Table 1 ijerph-19-16006-t001:** Lead concentrations in some food items (µg/g).

Food Samples	Mean	Range
Amaranthus	2.5 ± 0.029	1.9–2.9
Arrowroot	2.9 ± 0.071	2.2–3.6
Cowpea leaves	0.4 ± 0.04	0.0–1.6
Fruits (guava and mangoes)	0.3 ± 0.025	0.0–1.3
Solanum nigrum	0.3 ± 0.025	0.2
Tomatoes	2.1 ± 0.00	2.1
Beans	0.2 ± 0.29	0.173–0.230
Maize	0.1 ± 0.02	0.066–0.174

Adapted from: Makokha et al., 2008 [45].

**Table 2 ijerph-19-16006-t002:** Levels of lead in drinking water from different locations.

Sampling Locations	Lead Levels Values	Units	SANS Recommended Standard
School A	<0.01	mg/L Pb	<0.01
School B	<0.01	mg/L Pb	<0.01
River	<0.01	mg/L Pb	<0.01

Adapted from: Olufemi, Mji and Mukhola, 2019 [59].

## Data Availability

Not applicable.

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
