# Peer review of "Potential Health Risks of Lead Exposure from Early Life through Later Life: Implications for Public Health Education"

_ijerph, 2022, doi:10.3390/ijerph192316006_

Round 1

Reviewer 1 Report (Previous Reviewer 2)

Authors have now provided details about revisions and response to previous and remaining comments. English language has been revised too and overall the manuscript has improved.

No further comments. Minor edits can be corrected if accepted.

Spinger and Science Direct with first letter in uppercase in the Abstract and Methods (section 7). Also 'literature online databases' can be used in lieu of academic databases.

This manuscript is a resubmission of an earlier submission. The following is a list of the peer review reports and author responses from that submission.

Round 1

Reviewer 1 Report

The paper is improved by including more examples illustrating the problem of lead exposure in LMICs.  However, it still lacks organization and structure to create a compelling argument for the reader.  A shorter introduction highlighting the effects of lead exposure would be sufficient.  Then the authors could focus on the particular challenges of reducing lead exposures in LMICs and in low income communities in wealthier countries.  There are many typos and redundancies.

Author Response

Comment 1: The author has added a short introduction on the effects of lead under the section LMCIs Versus HICs.

Comment 2: The author has also added the challenges of reducing lead exposure in the LMICs and poor areas of wealthier countries or HICs still under the section LMICs versus HICs

Please note that the responses to the comments are highlighted in light green. Thanks

Reviewer 2 Report

The manuscript titled "Potential health risks of Lead exposure from earlylife through laterlife: Implications for Public Health Education" presents an interesting review about risk related to lead exposure with specific interest on children health.

Despite the interest, several issues should be addressed before it can be considered for publication.

The Introduction should be revised as some statements are very general and not helpful to the purpose of the study.

The sentence starting with “One of such heavy metals…” is not adequately worded as lead appears two times and there is no need to mention other heavy metals. The same for the subsequent sentence as the relevant information is that lead was used since Ancient time, leading to chronic exposure and related health problems (some of them should be mentioned as anemia, bone diseases, etc.).

The subsequent paragraph about pollution due to use of tetraethyl lead as anti-knocking agent is relevant but the concept is repeated several times. Rewording is recommended.

At the end of page 2, the issue is also related to the presence of old lead pipes not yet changed both in houses and water system supplies.

The examples about source of lead from foods and soil are interesting but limited to individual and specific cases, while the aim of the revision seemed to assess in general the issue of food, water and soil contamination. It is not clear why limiting the analysis to those studies. If the purpose was to assess the specific regional area (South Africa), please check if relevant studies are available. This is the most relevant concern about the review that does not implement a systematic search, but present results of studies selected with unclear and unspecified criteria, making very difficult to the reader to understand the rational for such selection.

The paragraph starting with “Some studies have also reported…” at page 4 is interesting but is a repetition of the use of lead in Ancient time and the link with the pollution in recent years has been already reported. A careful organization of the Introduction is highly recommended.

Table 1 does not add any useful information compared to the corresponding text, all values are below the limit which can be easily reported in the text. The main issue is to clarify if the value of 0.01 mg/L was the detection limit or lower values were found. The sentence starting with “Conversation with some of the…” is not scientifically sound, these are anecdotes which may be interesting but the relation between lead intake and health problem is only a far hypothesis, not the proven cause. Revision of the text is required.

Table 2 presents findings relating lead exposure and children health. The table is interesting but there is no indication of the criteria used for selection of those studies compared to several ones available in the literature (type of population, type of lead exposure, type of studies, type of outcome). The search should be performed according to the PECO statement (Population, Exposure, Comparison and Outcome) and more in general the entire study should aim at present results of a systematic search according to the PRISMA statement or similar guidelines for systematic review.

The comparison between Low and Middle income versus High income countries is valuable, but it is evident that some cited studies are missing from Table 2. Two comparison tables with studies divided by type of country (Low/Middle vs High) seems helpful to adequately present such important issue.

An additional issue is the lack of organization of the manuscript into sections like Results and Discussion. In the section 3, both findings from previous studies and authors’ thoughts and conclusions about the motivation of higher intake are included, while a clear division would help the reader in assessing on one hand the source of exposure from foods and role of nutrition, on the other hand the motivations of the higher intake in Low/Middle income countries.

Section 6 is interesting, but the supporting references are lacking. If the aim was to list the most important take home messages, a bullet point list of short sentences seems a better way. As reported, supporting citations must be included.

Author Response

Dear Reviewer

Thanks for your efforts in reviewing our manuscript. I will like to inform you that I have attended to the comments raised by the reviewer except for those ones that seem not to be compulsory. The answers are highlighted in purple colour and track changes. Find attached a copy of the revised manuscript. Thanks once again.

Round 2

Reviewer 1 Report

This paper is substantially improved, and is of much greater interest to the reader. The addition of new references and analysis of the continuing problem of lead poisoning in LMICs provides a unique and valuable perspective.  Well done.

The main remaining issues concern the organization of the topics, and the ways that different types of lead exposure are categorized.

For the Intro, it would be better to start off with a strong thesis, such as

"Lead has had many uses over millennia of human history.  Its malleability and resistance to corrosion made it useful for water pipes and vessels.  It also has numerous industrial uses, reflecting its unique chemical properties. The toxic effects of lead have been documented since ancient times (Waldron HA "Lead Poisoning in the ancient world.  Med Hist 1973.  17: 391-399), but lead poisoning was not well characterized until the Industrial Revolution, when occupational exposure to lead became more common, and began to affect workers and their families.  Despite growing awareness of the dangers of lead exposure, especially for children, lead continued to be used in plumbing, paints and gasoline, and a variety of household products, such as vinyl window blinds, glazed dishware, and toys through most of the 20th century. 

Give concise (about 2 paragraphs) explanation of consequences of lead exposure for both children and adults.  Emphasize that there is no safe level of lead exposure.

Give history of lead use being phased out in the US and other wealthy countries.  Introduction of lead monitoring, trends in blood lead levels in high income countries.  Note improvements in health and cognitive function that have been linked to these changes.  Mention continued problems in poor communities within these countries.

Give same info for LMICs.  When was lead removed from petro & paint?  Are blood lead levels monitored in children?  What are obstacles to preventing lead exposure in children? 

What are unique exposures in LMICs, such as informal sector mining and e-waste recycling, exposure of produce carried in open trucks along roads that have lead-containing dust, etc. 

[So basically say that lead is bad, we've known that it is bad for a long time, but have only relatively recently taken action to reduce lead exposures.  Wealthier countries were faster to remove lead-containing products from the market, and to monitor lead levels in children.  LMIC's have were slower to eliminate lead and do not have the resources to reduce exposures or monitor children's lead levels.  In addition, there are certain types of exposures that are more common in LMICs, and that are difficult to regulate (artisinal mining, e-waste recycling, open air tranport and marketing of food). 

Then continue with types of exposure.  Keep focus on LMIC's , but also mention that lead remains a challenge worldwide

A few things need to be clarified:

1)  You talk about lead in food.  What is the source of the lead?  Is it contaminated in the home by dust that contains lead (eg:  dust from the hands of children who eat without having washed their hands) or is it in the food before it enters the home (as part of the growing, processing or delivering of the food)?

2) The significance of lead in soil needs to be clarified.  To what extent does lead in soil contaminate food?  Talk about how lead in road dust can contaminate food & spices (eg:  when spices are dried on sunny stretches of roads).  I assume lead from soil also blows into homes and is carried in by shoes.

3) I would expect that soil lead concentrations near mines would be much higher than residual lead from petrol and paint - I think it should be in a different catagory, and could also include information about artisanal mining affecting workers and family members, as well as the general environment. 

I like the section on strategies to mitigate exposure.  This could be made more concise and clear. 

Overall, it would be best to have an editor go through and work on organization and avoiding redundancies.

Author Response

Dear Reviewer,

Thanks for your efforts in reviewing our manuscripts and for letting us know that the paper is now improved. God bless you real good. I will like to confirm that all the comments have been attended to and authors responses are given under each of the comments. Also the changes are effected on the manuscript and highlighted in red, Please find arttached a copy of the revised manuscript

REVIEWER 1 COMMENTS AND ANSWERS

Note: Please find the Authors answers below the reviewers’ comments

Reviewers’ comments:

  • This paper is substantially improved and is of much greater interest to the reader. The addition of new references and analysis of the continuing problem of lead poisoning in LMICs provides a unique and valuable perspective.  Well done.

Authors’ answer

  • Thanks for taking time to review our manuscript and for all the comments and for letting us know that the paper is improved. We really appreciate

Reviewers’ comments:

  • The main remaining issues concerning the organization of the topics, and the ways that different types of Lead exposure are categorized.

Authors’ answer:

  • With regards to this organization of the topics, we decided to adjust it a bit except if there is any suggestion. Also, the issue of the different types of Lead exposure has been addressed. For instance, the area that talk about Lead exposure from Paint and Gasoline has been given a subtitle.

Reviewers’ comments:

  • For the Intro, it would be better to start off with a strong thesis, such as

"Lead has had many uses over millennia of human history.  Its malleability and resistance to corrosion made it useful for water pipes and vessels.  It also has numerous industrial uses, reflecting its unique chemical properties. The toxic effects of lead have been documented since ancient times (Waldron HA "Lead Poisoning in the ancient world.  Med Hist 1973.  17: 391-399), but lead poisoning was not well characterized until the Industrial Revolution, when occupational exposure to lead became more common, and began to affect workers and their families.  Despite growing awareness of the dangers of lead exposure, especially for children, lead continued to be used in plumbing, paints and gasoline, and a variety of household products, such as vinyl window blinds, glazed dishware, and toys through most of the 20th century. 

Authors’ answer:

  • With regards to the intro part, the suggested statement above was considered but rephrased because we were not sure if it was a direct quotation from the author (Waldron 1973) or reviewers’ own idea or words. Some of the statements were used at the beginning of the introduction while the rest were used somewhere else

Reviewers’ comment

  • Give concise (about 2 paragraphs) explanation of consequences of Lead exposure for both children and adults.  Emphasize that there is no safe level of Lead exposure.

Authors’ answer

  • An explanation about the consequences of Lead exposure for both children and adults has been given but under separate headings with separate tables. The issue of no safe level of Lead exposure has also been emphasized.

Reviewers’ comment

  • Give history of Lead use being phased out in the US and other wealthy countries.  Introduction of Lead monitoring, trends in blood lead levels in high income countries.  Note improvements in health and cognitive function that have been linked to these changes.  Mention continued problems in poor communities within these countries.

Authors responses

  • The history about Lead use being phased out in the US and Australia given. Germany was also added. For LMICs, Nigeria was included
  • The issue of Lead monitoring and trends in BLLs in HICs also addressed
  • improvements in health and cognitive function that have been linked to these changes were also included.
  • Continued problems in poor communities within these countries were also described.

Reviewers’ comments

  • Give same info for LMICs.  When was lead removed from petro & paint?  Are blood lead levels monitored in children?  What are obstacles to preventing lead exposure in children? 

Authors’ answer

  • The same information was given for LMICs. Examples from Nigeria and South Africa and years of removal of Lead from petrol and paint. As well as obstacles, that prevent lead exposure in children

Reviewers’ comments

  • What are unique exposures in LMICs, such as informal sector mining and e-waste recycling, exposure of produce carried in open trucks along roads that have lead-containing dust, etc. 

Authors’ answer 

  • This was also addressed in the paper

Reviewers’ comments

[So basically, say that lead is bad, we've known that it is bad for a long time, but have only relatively recently taken action to reduce lead exposures.  Wealthier countries were faster to remove lead-containing products from the market, and to monitor lead levels in children.  LMIC's have were slower to eliminate lead and do not have the resources to reduce exposures or monitor children's lead levels.  In addition, there are certain types of exposures that are more common in LMICs, and that are difficult to regulate (artisinal mining, e-waste recycling, open air transport and marketing of food).

Then continue with types of exposure.  Keep focus on LMIC’s, but also mention that lead remains a challenge worldwide

Authors’ answer 

  • Apart from the common sources of Lead exposure already described in the paper, other sources of exposure that are more common in LMICs are reported and factors that make them to be difficult to regulate are given in the paper.

Reviewers’ comments

A few things need to be clarified:

1)  You talk about Lead in food.  What is the source of the lead?  Is it contaminated in the home by dust that contains lead (e.g.:  dust from the hands of children who eat without having washed their hands) or is it in the food before it enters the home (as part of the growing, processing or delivering of the food)?

Authors answers

  • The various sources of lead exposure from food and all the issues raised by the reviewers have been explained under the section Lead exposure in food

Reviewers’ comments

2) The significance of lead in soil needs to be clarified.  To what extent does lead in soil contaminate food?  Talk about how lead in road dust can contaminate food & spices (e.g.:  when spices are dried on sunny stretches of roads).  I assume lead from soil also blows into homes and is carried in by shoes.

Authors answers

  • These have also been addressed under the section

Reviewers’ comments

3) I would expect that soil lead concentrations near mines would be much higher than residual lead from petrol and paint - I think it should be in a different category and could also include information about artisanal mining affecting workers and family members, as well as the general environment. 

Authors answers

  • These have also been addressed in the paper

Reviewers’ comments

I like the section on strategies to mitigate exposure.  This could be made more concise and clearer. 

Authors answers

  • This has also been addressed in the paper

Reviewers’ comments

Overall, it would be best to have an editor go through and work on organization and avoiding redundancies.

Authors answers

  • The paper has been edited as suggested by the reviewer

                                                                                                                                        Submission Date

20 July 2022

Date of this review

26 Aug 2022 08:54:14

Reviewer 2 Report

Authors revised the Introduction taking into account the suggestions. An additional note about chronic diseases: cognitive impairment seems better than dementia as also children may be affected by lead poisoning, leading impairment of neurodevelopment.

Despite this revision, still methods for study selection and identification are not reported, both original articles (e.g. cross-sectional, cohort) and review are included. A clear indication on which studies are considered (population, type of exposure, design of the study) must be clarified. Some labels are not clear like empirical, while others are not correct, as for example Olufemi et al 2018 (actually 2019, check reference) is not an experimental study. A careful check and revision is recommended.

The manuscript writing must be improved, repetitions are present (e.g. industries at page 2, line 6) and several typos are present throughout the text. 

Authors should also prepare a point-to-point response for the previous comments, especially explaining why some were not addressed, providing a rebuttal letter which can clarify why some revisions were made and some not. Now only revised text (not in track-changes mode is provided, while revisions should be highlighted in some ways (e.g. red text). Please use also the previous comments and address them in a response letter.

Author Response

Dear Reviewer, 

Thanks for all your efforts in reviewing our manuscript. I have attended to all the comments raised and the responses are given under each of the reviewers comments. Also the changes are effected on the revised manuscript and highlighted in red color. (Please find the attached manuscript)

REVIEWERS 2 COMMENTS AND ANSWERS

Dear Reviewer,

Thanks very much for taking time to review our manuscript and thanks for all the comments raised.

Note: Please find answers to the comments below the reviewers’ comments

Reviewer’s comments

Authors revised the Introduction taking into account the suggestions. An additional note about chronic diseases: cognitive impairment seems better than dementia as also children may be affected by Lead poisoning, leading to impairment of neurodevelopment.

Author’s answers

  • This has been addressed in the manuscript
  • For example, with regards to the comment about dementia, I remembered it was suggested in the last review process that I should include dementia. Instead of removing it completely, I am adding cognitive impairment and impairment of neurodevelopment for children.

Reviewer’s comments

Despite this revision, still methods for study selection and identification are not reported, both original articles (e.g., cross-sectional, cohort) and review are included. A clear indication on which studies are considered (population, type of exposure, design of the study) must be clarified. Some labels are not clear like empirical, while others are not correct, as for example Olufemi et al 2018 (2019, check reference) is not an experimental study. A careful check and revision is recommended.

Author’s answers

  • The matter of methods for study selection and identification are now reported and this aspect is just before the heading “conclusion” in the paper
  • All the other issues raised here are addressed. Olufemi et al 2018 has been changed to Olufemi et al 2019
  • . I have also addressed the labels issue and types of study such as empirical for Zahran et al and experimental for Olufemi et al

Reviewer’s comments

The manuscript writing must be improved, repetitions are present (e.g., industries at page 2, line 6) and several typos are present throughout the text. 

Author’s answers

  • The matter of typos has been addressed in the paper and have also gone through editing

Reviewer’s comment

Authors should also prepare a point-to-point response for the previous comments, especially explaining why some were not addressed, providing a rebuttal letter which can clarify why some revisions were made and some not. Now only revised text (not in track-changes mode is provided, while revisions should be highlighted in some ways (e.g., red text). Please use also the previous comments and address them in a response letter.

Author’s answers

  • Most of the comments that were not addressed in the last reviewing process have now been addressed now. For example, the issue of method for study selection and identification.
  • The issue of health risk assessment was not addressed since this is just a report and no data collection. Another study is ongoing with data that will include health risk assessment.
